# Impact of Lipid Metabolism on Antitumor Immune Response

**DOI:** 10.3390/cancers14071850

**Published:** 2022-04-06

**Authors:** Nesrine Mabrouk, Baptiste Lecoeur, Ali Bettaieb, Catherine Paul, Frédérique Végran

**Affiliations:** 1Université de Bourgogne Franche-Comté, 21000 Dijon, France; nesrine.mabrouk@inserm.fr (N.M.); baptiste.lecoeur@u-bourgogne.fr (B.L.); ali.bettaieb@u-bourgogne.fr (A.B.); catherine.paul@u-bourgogne.fr (C.P.); 2LIIC, EA7269, University of Burgundy Franche-Comté, 21000 Dijon, France; 3Immunology and Immunotherapy of Cancer Laboratory, EPHE, PSL Research University, 75014 Paris, France; 4Team CAdIR, CRI INSERM UMR1231 “Lipids, Nutrition and Cancer”, 21000 Dijon, France; 5Centre Georges-François Leclerc, UNICANCER, 21000 Dijon, France; 6LipSTIC Labex, 21000 Dijon, France

**Keywords:** immune cells, lipid metabolism, cancer therapy, immunosuppression

## Abstract

**Simple Summary:**

One of the causes of failure of anticancer therapies is the reprogramming of lipid metabolism. Cells of innate and adaptive immunity present in the tumor microenvironment can be affected by this metabolic switch and thus present changes in their anti- or protumor phenotype. In this review, modifications induced by lipid metabolism will be described for innate immune cells, such as macrophages, dendritic cells and MDSCs, and also for adaptive immune cells, such as CD4+ and CD8+ T cells and B cells. Finally, antitumor therapeutic strategies targeting lipid metabolism will be presented.

**Abstract:**

Over the past decade, metabolic reprogramming has been defined as a hallmark of cancer. More recently, a large number of studies have demonstrated that metabolic reprogramming can modulate the differentiation and functions of immune cells, and thus modify the antitumor response. Increasing evidence suggests that modified energy metabolism could be responsible for the failure of antitumor immunity. Indeed, tumor-infiltrating immune cells play a key role in cancer, and metabolic switching in these cells has been shown to help determine their phenotype: tumor suppressive or immune suppressive. Recent studies in the field of immunometabolism focus on metabolic reprogramming in the tumor microenvironment (TME) by targeting innate and adaptive immune cells and their associated anti- or protumor phenotypes. In this review, we discuss the lipid metabolism of immune cells in the TME as well as the effects of lipids; finally, we expose the link between therapies and lipid metabolism.

## 1. Introduction

Immunosurveillance is a term used to describe the processes by which cells of the immune system look for and recognize not only foreign pathogens but also pre-cancerous and cancerous cells in the body. Schreiber and colleagues postulate that there is a continuous monitoring of our immune system in search of cancer cells [1]. This cancer immunosurveillance mechanism results from immune effector functions stimulated by recognition of stress ligands or antigens expressed on transformed cells [2]. It has been suggested and now demonstrated that immunosurveillance is composed of three phases [3]. In the first phase, called the elimination phase, tumor cells are killed by NKs and CD4+ and CD8+ cells. The second phase correspond to a state balance that is accompanied by a selection of tumor clones resistant to the immune system: this is the immune-editing phase. Finally, when the immune system becomes unable to destroy tumor cells, it is the third phase, called immune-escape. In this context, it is now accepted that the cells of the immune system are an integral part of the tumor microenvironment (TME). 

In the hallmarks of cancer described by Hanahan, one of the 14 biological capabilities acquired during tumor development is energy metabolism reprogramming [4]. Indeed, when tumors grow, the TME composition of nutrients, metabolites and cell types changes, and the cancer cells have to adapt to this new environment. Metabolic flexibility and plasticity during cancer progression has been widely described in cancer cells [5]. Thus, the extent to which an area is perfused seems to determine the preference for nutrients, i.e., glucose or lactose, for energy metabolism [6]. Other factors such as the heterogeneity found in every cancer, the cancer type and the presence of oncogenic mutations play a role in metabolic flexibility [4]. More recently, reprogramming of metabolism of immune cells has also been described as a new hallmark of cancer. These modifications to metabolism change the functionality of immune cells by controlling transcriptional and posttranscriptional events that are essential for the activation of immune cells [7]. Specific metabolic pathway alterations affect immune cell functions. The role of immune cells in cancer progression and the influence of metabolism on immune-cell function are the primary reasons for increased research efforts in the field of immune-metabolism. Increasing evidence suggests that energy metabolism could be responsible for the failure of antitumor immunity.

Innate and adaptive immune cells are involved in the TME and affect tumor growth by either triggering an inflammatory response to participate in the elimination of cancer cells or by displaying protumor immunosuppressive activity. Innate cells such as macrophages, dendritic cells (DCs), neutrophils, myeloid derived suppressor cells (MDSCs), natural killer cells (NKs), and innate lymphoid cells (ILCs) modulate the TME by controlling T-cell fate; thus they sculpt the TME but at the same time are sculpted by the TME [8]. For example, in physiological conditions, M1 proinflammatory macrophages promote both innate and adaptive immunity. However, in a tumor microenvironment these cells rapidly acquire the M2 phenotype, promoting tumor progression [9]. Among adaptive immune cells, T cells can be found in tumors with various phenotypic subpopulations (CD4+ and CD8+) and functional (effector, memory) and differentiation (CD4+ Th1, Th2, Th17 and Treg) states. These different cells have been described as being able to modulate tumor growth either by direct engagement or by stimulating other cells in the tumor microenvironment [10]. A large number of studies have demonstrated that metabolic reprogramming of immune cells can modulate their differentiation and functions, and thus modify the antitumor response. In this review, we discuss the lipid metabolism of immune cells in the TME as well as the effect of lipids in modulating the immune response. Finally, we expose the link between therapies and lipids.

## 2. Lipid Metabolism in Myeloid Cells

Myeloid cells belong to innate immunity and develop from a common myeloid progenitor cell derived from multipotent hematopoietic stem cells. These innate cells include dendritic cells, monocytes, macrophages, neutrophils, basophils, eosinophils, erythrocytes, myeloid-derived suppressor cells (MDSCs) and megakaryocytes (that become platelets). These cells are implicated in trauma, inflammatory and autoimmune diseases, infections, graft-versus-host diseases and cancer [11,12,13]. The various states of polarization, activation and regulation of myeloid cells are accompanied by adaptation of lipid metabolism [14,15]. In contract, lipid metabolism modulates the activity of these cells [16].

### 2.1. Lipid Metabolism in Macrophages

Macrophages belong to myeloid cells that are found in almost every tissue in the body, where they act as sentinels of the innate immune system [17]. Two different subtypes can be defined based on their activation program. M1 macrophages are classically activated. They have proinflammatory features and occur in response to bacterial infection products (e.g., lipopolysaccharide (LPS) and interferon gamma (IFN**γ**)). This subtype is also known for its antitumor and proinflammatory properties. M2 macrophages, alternatively, are activated in response to products associated with parasitic infection (e.g., Schistosoma egg antigen and interleukins (IL)-4 and -13) and have antiparasitic activities and tissue repair and remodeling features [17,18]. This second subtype exhibits protumor and anti-inflammatory activities. Experimental data suggest tumor associated macrophages (TAMs) to be largely biased towards the M2 phenotype [19,20]. The metabolic reprogramming of M1 towards M2 macrophages primarily relies on oxidative phosphorylation (OXPHOS) [21]. Several other factors can stimulate the polarization of macrophages towards M2 cells, including AMP-activated protein kinase (AMPK), an enzyme that plays a crucial role in cholesterol, glucose and fatty acid (FA) uptake and oxidation. Indeed, it has been shown that, opposite to M1 macrophages, FA uptake and oxidation is upregulated in the IL-4-induced M2 phenotype [22]. Besides AMPK, other signals and pathways can also influence macrophage polarization into the M2 phenotype, including mTOR complex 2 (responsible for lipid metabolism and biosynthesis), triacylglycerol (essential for fatty acid oxidation (FAO)), arachidonic acid metabolism at a low level of COX2 and a high level of COX1 and microsomal isoforms of PGE synthase (mPGES), and peroxisome proliferator-activated receptors (PPARs) (a key sensor of lipids) [23]. In fact, it has been demonstrated that the blockade of triacylglycerol substrate uptake (via CD36 deletion) of free fatty acids (FFAs) release (via lysosomal lipolysis alteration) induces impairment of both OXPHOS and IL-4-induced M2 gene expression by murine macrophages [24]. In fact, when macrophages are cocultured with tumor cells or isolated from the tumor bed, it has been shown that they accumulate lipids and use the CD36 and LOX-1 scavenger receptors to transport FAs from the TME [24,25,26]. Additionally, FAO inhibition with etomoxir or CD36 deletion in TAMs slows tumor growth and M2 polarization [26]. Arachidonic acid (AA) displays different characteristics depending on whether it is utilized by M1 or M2 macrophages. In fact, as opposed to M1 macrophages, IL-4-induced M2 polarization is associated with upregulation of arachidonate 15-lipoxygenase (ALOX5) and COX1 and downregulation of mPGES and COX2 expression [22]. Effectively, it has been demonstrated that LPS induces upregulation of mPGES, and that contrasting changes are found in response to IL-4 and IL-13 [22,27]. PPARs (α, δ and γ) are ligand-activated transcription factors belonging to the nuclear receptor superfamily and that control and regulate enzymes involved in transport, synthesis, storage, mobilization, activation and oxidation of FA [17]. PPARδ, for example, has been found to be essential for M2 differentiation [28,29,30]. Thomas et al. also revealed that PPAR expression is upregulated in M2 macrophages recovered from mice exposed to parasitic helminths. Additionally, these cells overexpress an eicosanoid ligand for PPARδ [31].

Recently, it has been demonstrated that lipids can also play an important role in TAM generation and function in the tumor microenvironment. These scientists revealed an increase of CD36 level and accumulated lipids in TAMs when compared to control macrophages. In addition, these cells use FAO instead on glycolysis for energy, which promotes several signaling pathways leading to the transcription of genes implicated in TAM generation and function [26]. It has been demonstrated that PPARβ/δ is also implicated in TAM’s acquisition of immune-suppressive activity. Indeed, in TAMS, once PPARβ/δ is activated by M-CSF derived from tumors, it drives proangiogenic (VEGF) and immunosuppressive (IL-10, arginase-1) gene expression [32]. 

Concerning M1 macrophage reprogramming, it seems that it is more reliant on glycolysis and is characterized by increased reactive oxygen species (ROS), nitric oxide and prostaglandin (PG) production [21]. Moreover, it has been shown that lipid metabolism also has several effects on inflammation induced by TLR agonists. As a matter of fact, an increase in the concentration of desmosterol, a cholesterol precursor, is observed in M1 macrophages. Moreover, in mice administered a high-fat diet, the inhibition of DHCR24 (a catalyzer of desmosterol synthesis) induced an upregulation of M1-activation-gene expression [30,33].

All of these studies have been performed on mice, however—little is known about the metabolic reprogramming of human macrophages, and most research has shown a controversial aspect of this immune-cell subtype in human. This concerns iNOS, NO and arginase expression but also glycolysis and OXPHOS metabolic reprogramming [34]. As a matter of fact, recently, Vijayan V. et al. demonstrated that, contrary to mouse bone marrow-derived macrophages, human, LPS-activated, peripheral-blood, monocyte-derived macrophages rely mostly on OXPHOS rather than glycolysis for the generation of ATP [21]. 

### 2.2. Lipid Metabolism in Dendritic Cells

Dendritic cells (DCs) are potent antigen-presenting cells that link the innate with the adaptive immune system. Lipids have been shown to modulate immune responses and to induce negative effects on DC activity [35]. Oxidized lipid accumulation in tumor-infiltrating DCs is associated with antigen cross-presentation function blockade and MHC class II orientation, since DC activity was restored in response to acetyl-CoA carboxylase inhibition [35,36]. Zeyda et al. revealed that n-3 and n-6 polyunsaturated fatty acids (PUFAs), such as arachidonic and eicosapentaenoic acid, alter DC differentiation and activation, leading to inhibition of their efficacy in inducing T-cell-mediated immunity [37]. Similarly, the results of Weatherill et al. showed that docosahexanaenoic acid, an n-3 PUFA, induces a decrease in CD40, CD80 and CD86 costimulatory molecule expression and MHC II and cytokine secretion. Contrasting changes were highlighted in response to lauric acid, a saturated FA. Hence, docosahexaenoic acid was associated with upregulated T-cell activation, whereas the opposite was observed in response to lauric acid [38]. Moreover, Aliberti et al. demonstrated that lipoxin A4 (LXA4), a metabolite of AA, induces downregulation of IL-12 responsiveness of DCs after microbial stimulation in vivo [39]. Furthermore, Shamshiev et al. reported diminished IL-2, -6 and TFN-α production by CD8α DCs isolated from dyslipidemic mice; these cells also favored Th2 cell polarization in vitro [40]. Lipid accumulation in DCs is one of the main mechanisms of DC dysfunction in tumors. It can reduce antigen-processing capacity, downregulate CD86 expression and overexpress the tolerogenic cytokine IL-10 [41]. In ovarian cancers, the expression of fatty acid synthase (FAS), the key enzyme in de novo lipogenesis, is increased. FAS increases FA synthesis in these cancer cells, and the elevated FA concentration in the TME leads to FA accumulation in DCs, thereby affecting their function [42]. Other studies have shown that the accumulation of oxidized lipids, particularly triacylglycerol (TAG), can cause DC dysfunction. The accumulation of lipid droplets in ovarian cancers is also responsible for the loss of ability of DCs to induce an antitumor response [43]. In melanoma, DCs—through the wnt5a-β-catenin-PPARg signaling pathway—can upregulate the expression of FA transport protein carnitine palmitoyl transferase-1a (CPT1A), driving the FAO process and protumor effects [44]. DCs in tumors, through oxidized lipid accumulation, can inhibit T-cell function and promote tumor progression by activating the endoplasmic reticulum stress response via the IRE1 protein [45].

### 2.3. Lipid Metabolism in MDSC

Lipids play a crucial role in myeloid-derived-suppressor-cell (MDSC) activity, which is vital for maintaining homeostasis, cell-membrane integrity, signaling and healthy performance [46,47]. In fact, it has been demonstrated that MDSCs recovered from cancer patients or tumor-bearing mice have higher lipid accumulation compared to controls [48,49] and thus have decreased T-cell activity, leading to immune suppression [50]. Indeed, it has been shown that in the tumor microenvironment, oxidized lipids are considered an important energy source for MDSCs’ immunosuppressive role. There is evidence that CD8+ T cells have their activity drastically suppressed by MDSCs with high lipid overload as opposed to MDSCs with normal lipid content. This lipid accumulation can be associated with an elevated FA uptake, as demonstrated by Cao et al., who showed that upregulation of fatty acid transport protein 4 (FATP4) expression was observed in tumor-derived MDSCs [51]. MDSCs can be divided into two different subtypes: monocytic (M) and polymorphonuclear (PMN). The latter have been demonstrated to increase the expression of fatty acid transport protein 2 (FATP2) in human and mouse polymorphonuclear models. This was correlated with the acquisition of an immunosuppressive function by these MDSCs and cancer development. In fact, when the researchers selectively inhibited FATP2, an abrogation of PMN-MDSC activity as well as a delayed tumor progression was observed. Veglia et al. revealed that the mechanism by which this transporter protein works involves the accumulation of AA, a key precursor of prostaglandin E2 (PGE2), which is responsible for CD8+ T-cell inhibition. This is the result of neutrophils reprogramming to PMN-MDSCs due to signal transducer and activator of transcription 5 (STAT5) signaling pathway activation, induced by GM-CSF and leading to FATP2 expression (Figure 1) [49]. STAT3 signaling induced by G-CSF is another pathway involved in MDSC expansion, differentiation and activation [52]. STAT3 and 5 have been reported to play an important role in neutral lipid accumulation in MDSCs. In fact, Al-Khami et al. revealed that STAT3 and STAT5 inhibition in MDSCs, using FLLL32 and pimozide, respectively, reduced intracellular neutral lipid accumulation, which is associated with a decrease of MDSC immunosuppression activity driven by arginase-1, iNOS and PGE2 expression [48]. PGE2 represents a fundamental bioactive lipid produced in response to cyclooxygenase-2 (COX-2). COX-2 induces AA transformation to an unstable intermediate, prostaglandin G2 (PGG2), then to endoperoxide H2 (PGH2) and later into PGE2 (Figure 1) or any of four other prostanoids (TXA2, PGD2, PGF2a and PGI2) [53]. COX-2 is also implicated in increasing MDSC proliferation, which is correlated with an upregulation of arginase-1 and iNOS expression in murine tumor-infiltrating leukocytes [54], thereby stimulating cancer cell proliferation. PMN-MDSCs also overexpress oxidized low-density receptor 1 (ORL1/LOX-1) [48]. Endoplasmic reticulum stress appears to be responsible for LOX-1 expression in PMN-MDSCs [55,56]. Although the role of LOX-1 in lipid metabolism and the suppressive activity of PMN-MDSCs is not yet fully explored, LOX-1 has been shown to be associated with an immunosuppressive gene signature [48]. 

In the tumor microenvironment, MDSCs can be found taking up FA, which they can utilize via several pathways. Yan et al., in 2013, demonstrated that PUFAs enhance both the accumulation and the differentiation of PMN-MDSCs in vitro and in vivo (in mice administrated with PUFA-enriched diets). These cells have a more potent immune-suppressive activity when compared to control PMN-MDSCs, suggesting that the anti-inflammatory property of PUFAs is also due to their action on myeloid suppressor cells. This group also demonstrated that the JAK–STAT3 signaling pathway is involved in the mechanism of action of PUFAs [13]. Another group of researchers indicated that culturing a myeloid-suppressor cell line (MSC-2) in the presence of sodium oleate, a long-chain, unsaturated FA, induced an increased lipid droplet accumulation and thereby upregulated their immune-suppression function [57]. Moreover, Cao et al. found that treatment of MDSCs with linoleic acid, another unsaturated FA, makes them more immunosuppressive than those treated with palmitic acid [51].

PPARγ was also highlighted to play a critical role in MDSC neutral-lipid-metabolism signaling. Indeed, increased PPARγ activity impairs MDSC-mediated proliferation and leads to tumor proliferation and metastasis [58]. Interestingly, mouse and human tumor-infiltrating M- and PMN-MDSCs show a preferential elevation of FA uptake and FAO over glycolysis, suggesting that the immunosuppressive activity of these cells inside the tumor is dictated by this metabolic profile [50]. Further, Park J. et al. demonstrated that in a model of Lewis lung cancer, M-CSF produced by cancer cells increased endogenous FA synthase, which is responsible for PPARβ/δ activation in tumor myeloid cells, leading to an increase of their immunosuppressive function. Indeed, PPARβ/δ is known to induce immunosuppressive and proangiogenic gene expression (such as IL-10, Arg1 and VEGF), generating tumor escape and progression [59]. Another molecular link between myeloid-cell development, activity and metabolism is retinoid X receptor (RXR), which is a cellular sensor of lipid hormones and metabolites. Hence, inhibition of subunit α of this receptor is crucial for the complete differentiation of human myeloid progenitor cells to neutrophils [32,60]. Moreover, it has been demonstrated that all-trans retinoic acid (ATRA), an agonist of RXR, plays a vital role in the differentiation of MDSCs into macrophages and DCs [32,61].

## 3. Lipid Metabolism in Adaptive Immunity

Adaptive immunity is the second line of defense of organisms. It is triggered when innate immunity is no longer able to overcome aggression. It is now accepted that adaptive immunity has an essential place in the fight against but also in the development of cancers. The link between lipid metabolism and the functional activity of T and B cells is not yet fully elucidated, but several studies suggest that the differentiation of these cells depends in part on extrinsic and intrinsic lipid signals. The quantity and quality of FAs in the extracellular environment are decisive in deciding cell fate.

### 3.1. Role of Lipid Metabolism in T-Cell Activation

After TCR and CD28 costimulatory signals, metabolic activation occurs in order to sustain the subsequent proliferation and differentiation of naïve T cells. The metabolic requirement for CD4 effector T cells (Teff) Th1, Th2, Th9, Th17 and Tfh is mainly supplied through an aerobic glycolysis process, as in cytotoxic CD8+ cells, while regulatory CD4+ T cells (Tregs) or exhausted CD8+ cells mainly rely on FAO [62]. Since phospholipids and cholesterol are the main building stocks of the plasma membrane, the metabolism switch from FAO to fatty acid synthesis (FAS), occurring after T-cell activation, suggests that de novo lipid synthesis actively sustains Teff expansion [63]. Consistent with this, FAS suppression in Teff cells after acetyl-CoA carboxylase (ACC) inhibition, a key enzyme involved in the first step of de novo lipid synthesis, broadly impairs their proliferation [64]. Glucose fueling FAS in Teff is assured by an increase in the glucose transporter GLUT1 expression, mediated by mTORC1, after TCR activation. This has been shown to be critical for T-cell proliferation and effector function [65]. The metabolic changes that occur in T cells following TCR triggering and CD28 costimulatory signaling are mediated through PI3K and Akt pathway signals, which activate the mTOR axis [66].

The mTOR pathway is divided into two distinct complexes: mTORC1, known to be inhibited by rapamycin, and mTORC2, which rapamycin can only inhibit under specific conditions [67]. These two complexes orchestrate a distinct manner of expression of numerous enzymes and transcription factors involved in both catabolic and anabolic processes [68]. As a result, they actively regulate lipid and glycolytic metabolism as well as protein synthesis and autophagy [69]. In immune cells, the role of mTORC1 in lipid metabolism is now well-characterized, while that of mTORC2 is still poorly understood apart from its implication in the de novo synthesis of sphingolipids [70]. During T-cell activation, mTORC1 is responsible for the switch from FAO to FAS across a TCR–mTORC1–SREBP (sterol-regulatory-element binding protein) signaling axis [71] as it promotes FA uptake through a TCR-mTORC1–PPARδ pathway [72]. In this context, the direct pharmacological inhibition of mTORC1 or the deletion of its associated protein, raptor (regulatory protein associated with mTOR), causes a significant reduction in de novo lipid synthesis after T-cell activation [73]. Moreover, and despite the promotion of FAS, mTORC1 appears to be essential for Treg suppressive function [74]. The mTORC1 pathway is mainly regulated by the AMPK enzyme, which acts as an intracellular energy sensor whose activation occurs when intracellular ATP levels drop [75]. When activated, AMPK inhibits the mTOR pathway by phosphorylating mTORC1 and, consequently, abrogates glycolysis and FAS to promote FAO [66]. These two opposing signaling pathways therefore play a crucial role in metabolic reprogramming following TCR activation in T cells, and mainly orchestrate the metabolic distinction between effector and regulatory T cells. In this way, FA metabolism complements glycolysis in the selective regulatory-T-cell expansion during tumor growth.

### 3.2. Influence of Lipid Metabolism on T-Cell Differentiation and Function

Although T-cell differentiation is mostly dictated by the cytokine environment, both FFA and triglyceride (TGs) content in the extracellular environment can influence this process by promoting the generation of a specific T-helper subset. This is particularly described for Th17 and Treg subtypes, the differentiation of which is particularly affected by these parameters. Th17 cells are already known for their plasticity and ability to switch toward Th1-like or Treg-like phenotype depending on the extracellular environment [76], and, recently, it appeared that glycolysis and FA synthesis are also implicated in this process [77]. Like other effector T cells, the Th17 subset primarily uses glycolysis, and glucose deprivation or Rapamycin treatment drastically reduces their proliferation [78]. However, Th17 cells still present reliance on FAS that was found to be more pronounced than in Th1 cells but not as much as in Treg cells [77]. Consistent with the need for FAS by Th17 cells, activation of the AMPK pathway in vitro and in vivo results in a decrease of the Th17 cell population in favor of Tregs by promoting FAO and shutting down FAS [79]. Additionally, Rapamycin-induced mTOR inhibition appears to promote the Treg population in mice by facilitating FAO while inversely abrogating glycolysis in Th17 cells [80]. Consistent with this, FAS disruption mediated by ACC inhibition was also shown to restrain Th17 cell generation and to promote Treg development, indicating that Tregs rely on the uptake of exogenous FFAs [64]. Conversely, obesity in humans preferentially allows Th17-cell differentiation by inducing ACC1 expression, which in turn increases the production of retinoid-related orphan receptor gamma t (RORgt) ligand and thereby its binding to the *il17a* promoter [81].

By taking all the effector T cells together, lipid metabolism can affect the balance between Teff/Treg cell populations on a larger scale. It has been shown that Tregs express a high amount of both isoforms of diacylglycerol acyl transferase 1 and 2 enzymes (DGAT1 and DGAT2), whose role is to esterify FFA on diacyl glycerol [82]. DGAT1 inhibition in Tregs leads to an increased level of active protein kinase C (PKC) and nuclear factor-kappa B (NFκB), while *Foxp3* expression is impaired. On the other hand, a study has demonstrated that DGAT1 expression in Teff is linked to a reduction in Treg frequency, thus promoting the establishment of a proinflammatory environment during autoimmune encephalomyelitis (EAE) [83]. In this way, DGAT1 seems to play an opposite role in Treg stability and formation and could selectively be a target of Teff or Treg depending on the pathological condition. 

It is now well known that Treg metabolism differs from Teff by being much more dependent on FAO than glycolysis for its survival and suppressive functions [62]. *Foxp3* expression has been shown to be sufficient to reprogram Treg metabolism by upregulating both expression of FAO and mitochondrial oxidative phosphorylation (OXPHOS) proteins and enzymes [84]. FOXP3 also dampens the aerobic glycolysis process in Tregs by suppressing MYC expression [85], making Treg more resistant to low-glucose and high-lactate environments [86]. Therefore, *Foxp3* deletion in Tregs drives a reconversion of their metabolism to a metabolic program similar to Teff. A recent study has shown that this reprograming partially depends on mTORC2 unveiling, which limits the lipid metabolism of Tregs. While *Foxp3* deletion in Tregs induces a metabolic program similar to that of Teff, inhibition of mTORC2 by *Rictor* deletion partially restores FAO in Tregs, indicating that FOXP3 may orchestrate Treg lipid metabolism partly by regulating the mTORC2 axis [87]. Nevertheless, the underlying mechanism remains to be elucidated. Tregs proliferating into the tumor microenvironment (TME) accumulate intracellular lipids under the intrinsic control of PPARδ [88]. This latter δ is known to drive CD36 scavenger receptor expression and FA uptake in response to lipid ligands, then feeding FAO-driven OXPHOS in Tregs. Additionally, the Treg lipogenic program could also be under the control of the OX40/OX40L signal that triggers TRAF6 activation, which is essential for lipid metabolism in memory T cells [89]. Treg homeostasis is therefore subject to the lipid content in the extracellular environment but also to its composition. For example, oleic acid has recently been reported to amplify FAO-driven OXPHOS metabolism in Tregs, thus strengthening *Foxp3* expression and stabilizing Treg lineage and immune-suppressive functions. In addition, when lipid uptake is impeded following CD36 deletion, it has been observed that tumor-infiltrating Tregs are specifically impaired. Their frequency is reduced, and their suppressive functions are blunted while they produce a heightened quantity of IFNδ and TNFα. Inhibition of sterol regulatory element-binding protein (SREBP)-dependent lipid synthesis and metabolic signaling in Tregs triggers effective antitumor immune responses. SREPB activity is upregulated in intratumor Tregs, and SCAP deletion (factor responsible for SREBP activity) inhibits tumor growth and stimulates an antitumor immunotherapy response. SCAP/SREBP signaling coordinates lipid synthesis and inhibitor-receptor signaling in Tregs, making it a new target to reduce Treg activity in tumors [90] (Figure 2). 

### 3.3. Fatty-Acid Oxidation Influences Memory and Exhaustion of T Cells

After the initial immune response, a small percentage of T cells will persist as long-lived memory T cells divided into three subsets: central memory (Tcm), effector memory (Tem) and tissue-resident memory (Trm) T cells. They exhibit a quiescent and non-proliferative state [91]. Therefore, and unlike Teff cells, they display a catabolic metabolism relatively close to naïve T cells, mainly relying on both glucose and FAO [92]. However, memory T cells are distinguished from naïve T cells as both CD4+ and CD8+ memory T cells display a heightened spare respiratory capacity (SRC), mitochondrial content and expression of proteins belonging to the electronic transporter chain (ETC), and therefore promote OXPHOS [93,94]. FAO has been described as essential for memory CD8+ cell generation by the demonstration that *TRAF6* deletion in mice greatly impeded their generation as the result of altering gene expression linked to FA metabolism [89]. This result was supported after demonstrating that, beyond the usual TRAF6 pathway, TRAF6 also activates the AMPK pathway in CD8+ T cells, thus promoting FAO and memory CD8+ generation [95] Memory CD8+ cells exhibit lower FA uptake than CD8+ effector T cells and mainly fuel FAO through FA and TG synthesis, using glucose as a raw material [96]. A recent paper associated lipid accumulation in TME CD8+ cells with increased expression of CD36. This is also correlated to the progressive dysfunction of T cells. The use of CD36-deficient models has made it possible to show that CD36-deficient T CD8+ cells retain their effector functions in TME compared to wildtype CD8 cells. CD36 promotes the uptake of oxidized low-density lipoprotein (OxLDL) by T cells, which induces activation of the p38 pathway [97]. Recently, a team has identified in the blood and tumors of melanoma patients a new CD8+ subpopulation with high levels of OXPHOS, strong expression of the ectonucleotidases CD38 and CD39, exhaustion markers and cytotoxicity. A high proportion of this CD8+ subpopulation correlates to immunotherapy resistance [98] (Figure 3). 

### 3.4. Influence of Lipid Metabolism on B Cells

B cells are the major actors of the humoral response through the secretion of antigen-specific antibodies (Ab); they also actively support T-cell activation. Few studies have focused on understanding B lymphocyte metabolism compared to other immune cell types, and their metabolic demands at different stages of their development remains poorly understood.

Naïve B cells have very little metabolic activity. Indeed, before their activation, B cells have a low number of mitochondria [99] and depend mostly on FA metabolism. Following their stimulation, they increase oxidative phosphorylation. It has been shown that B cells harbor a strong expression of CD36, an FA translocase [100], which is correlated with a strong propensity of B-lymphocytes to uptake linoleic acid. This increases the affinity for lipid metabolites via upregulation of lipids. Thus, CD36 scavengers may affect B-cell activity. Activated B cells necessarily require high energetic supply and protein synthesis to properly engage rapid proliferation and Ab secretion. While quiescent naive B cells appear to rely on FA as a major source of energy [101], previous studies have shown that T-independent stimulation with LPS induces a remodeling of their metabolic program by increasing OXPHOS, the tricarboxylic acid cycle and nucleotide synthesis, but not glycolysis, for which they still show dependency.

Recently, a new player in B-cell metabolism has been identified. Sirtuins (SIRT) are nicotinamide adenine dinucleotide (NAD+)-dependent deacetylases. Regarding β-oxidation, it seems that members of the SIRT family do not have the same effect. SIRT1 increases the activity of β-oxidation by activating both PPARα and PGC1α, thereby promoting the expression of target genes linked to increased lipid utilization. At the same time, SIRT1 inhibits lipid synthesis by deacetylating SREBP-1c or by suppressing PPARγ. In case of caloric restriction, SIRT3 activates long chain acyl-CoA dehydrogenase (LCAD), promoting β-oxidation. On the other hand, SIRT4 inhibits the transcription of genes underlying β-oxidation, such as PPARα, while SIRT6 represses the transcription of genes linked to FA synthesis [102]. The role of SIRT proteins in B lymphocytes has been described from a pathological perspective. SIRT1, instead, delays the onset of autoimmunity in mice. SIRT1-deficient mice have immune complexes deposited in their livers and kidneys, suggesting an autoimmune disease due to the absence of SIRT1 [103]. Another study showed that SIRT1 and SIRT2 contribute to the pathogenicity of chronic lymphocytic leukemia (CLL). SIRT1 mRNA and protein expression were increased in B cells derived from human patients with CLL. In addition, pharmacological inhibition of SIRT1 and 2 using inhibitors EX-527 and sirtinol in peripheral blood mononuclear cells (PBMC) from patients with CLL caused cytotoxicity [104]. These results indicate that SIRTs are potential targets for patients with CLL or other diseases related to B cells

## 4. Targeting Lipid Metabolism as a Therapeutic Strategy

### 4.1. Limiting Macrophage Pro-Tumor Effect

PPARγ regulates the production of anti-inflammatory cytokines by macrophages polarized to M2 [24]. Administration of the PPARγ agonist rosiglitazone increases the cytotoxicity of CD8+ T cells, in particular by modifying their effect on myeloid cells [105]. Other interesting targets of lipid metabolism may reside in the COX2–mPGES1–PGE2 axis. In fact, overexpression of COX2 and PGE synthase-1 by TAMs promotes AA production and conversion to PGE2, which is directly associated with increased expression of PD-L1 by TAMs [106]. Conversely, another study found that upregulation of FAO in lipid-laden foam cells inhibits the accumulation of lipids (TG) and the production of proinflammatory cytokines, and reduces damage induced by ROS in macrophages. Thus, the induction of FAO in foam cells may have therapeutic potential for the treatment of chronic inflammation [15,107]. In addition, Di Biase et al. reported the beneficial effects of fasting or low-calorie diets on improving the antitumor effect of chemotherapies in preclinical models of melanoma and breast cancer. This effect could be linked to an increase in CD8 cytotoxicity against tumor cells [108]. As previously published, drugs such as ATRA, an RXR agonist, attenuate the immunosuppressive nature of the tumor microenvironment. Therefore, combining these drugs with chemotherapy or immunotherapy could represent a new strategy to prevent cancer progression. Mirza et al. also demonstrated that ATRA was able to eliminate immature myeloid suppressor cells and improve the myeloid/lymphoid ratio, DC function and antigen-specific T-cell response in patients with metastatic kidney cancer. Thus, ATRA in combination with cancer vaccines could be a promising approach to improve the immune response by boosting DCs and thus eliminate cancer [109]. Lipid metabolism plays an essential role in the polarization and function of TAMs in vivo and in vitro; thus, targeting FAO in TAMS could be a potent therapeutic modality to improve the efficacy of anticancer therapies [26]. A relationship between cholesterol biosynthesis and the stimulator of interferon genes (STING)-dependent IFN type I response has been observed in macrophages and DCs [110]. The results suggest that disruption of cholesterol biosynthesis may enhance the beneficial effect of STING agonists in cancer immunotherapy. Thus, treatment with statins showed antitumor activity synergistic with the administration of IL-2 in the activation of NK cells in a myeloid-cell-dependent manner. Statins associated with IL-2 increased the production of IL-1β and IL-18 by macrophages and DCs, which slowed tumor progression. These results suggest an essential role of cholesterol metabolism in the function of myeloid cells (Figure 4).

### 4.2. Diminishing MDSC Immunosuppression

Antibody-based immunotherapies blocking checkpoint inhibitors (CPI) are effective in only 10–15% of patients. MDSCs present in the tumor bed are among the factors that may be responsible for these treatment failures. Inhibition of the immunosuppressive functions of MDSCs seems to be a prerequisite for success of immunotherapies. Different strategies have been studied.

In MDSCs, activation of PPARγ attenuates immunosuppression [58]. Furthermore, in a model of pancreatic cancer, the combination of gemcitabine and rosiglitazone decreases the immunosuppression induced by MDSCs and restores the efficiency of CD8+ T cells [111].

Other interesting targets of lipid metabolism may reside in the COX2/mPGES1/PGE2 axis. Indeed, the overexpression of COX2 and PGE2 synthase-1 by MDSCs promotes AA production and conversion to PGE2, associated with increased expression of PD-L1 by MDSCs [106]. In mouse models of lung (LLC1) and colon (CT26) cancer, studies have shown that certain tumor-derived factors may increase FA uptake via FATP2 and subsequent release of PGE2 by PMN-MDSC. Indeed, FATP2, induced by GM-CSF, regulates the accumulation of lipids and ROS in MDSCs [112]. This is directly correlated with the immunosuppression of CD8+ T cells. Administration of Lipoferms, an inhibitor of FATP2, reduces tumor progression in various cancer models. In addition, combined treatments using Lipoferms with anti-CTLA4 or anti-PD-L1 strongly inhibit tumor growth [49,112]. The combination with anti-PD-1 has similar effects in the TC-1 lung tumor model. These beneficial effects appear to be related to the reduced release of PGE2 by PMN-MDSC. In combination with CSF1R-blocking antibodies, Lipoferms produced antitumor effects in the LLC1 model, suggesting the possible involvement of TAMs [49,113]. As discussed previously, FATP2 plays an important role in reprogramming neutrophils into PMN-MDSC and promoting cancer progression. Therefore, inhibition of FATP2 may be able to reduce the immunosuppressive activity of PMN-MDSC and thus act as a therapeutic target to fight cancer [49].

In another approach, adoptive transfer of T cells in combination with etomoxir, an inhibitor of CTP1, an FAO enzyme, significantly reduced tumor progression in the LLC1 model compared to adoptive transfer alone or etomoxir alone. In this experiment, infiltration of the transferred cells into the tumor microenvironment seems to be increased and associated with a strong production of IFNγ. On the other hand, etomoxir does not seem to change the number of MDSCs, but does affect their function, resulting in a decrease in the expression of ARG1 as well as the cytokines involved in their expansion (G-CSF, GM-CFS, IL-6, IL-10). Thus, it seems that modulation of FAO can considerably increase the effectiveness of chemotherapy by targeting immunosuppression [50]. Other studies indicate that etomoxir promotes the polarization of inflammatory macrophages [114].

Medicines such as metformin, used in the treatment of type II diabetes, are capable of inhibiting complex I of the mitochondrial respiratory chain, which is essential during FAO [115]. Various studies have shown that metformin has the ability to limit the accumulation of MDSCs and their immunosuppressive functions [116,117]. Despite the few studies available on the effects induced by cholesterol metabolism and immunotherapy, emerging evidence shows a potential benefit in targeting cholesterol synthesis as well as its transport in different types of immune cells [118]. An LXR agonist (RGX-104) induces the regression of several cancers by reducing tumor MDSC infiltration. In particular, Tavazoie et al. have demonstrated that coadministration of LXR agonists with adoptive transfer of CTL increases the antitumor activity of transferred CTL and survival of mice. At the same time, the combination of the LXR agonist and anti-PD-1 significantly reduces MDSCs and tumor growth [119].

Another potential target for FA-regulated MDSC functions is the STAT family. A decrease in lipid accumulation, mitochondrial metabolism and MDSC activity has been reported in response to pharmacological inhibition of STAT3 and STAT5 [13]. In addition, the use of the JAK inhibitor JSI-124 induces negative regulation of STAT3 phosphorylation, leading to a decrease in the effect of PUFAs on MDSCs [120]. Additionally, Hossain et al. revealed upregulation of FA and activation of FAO incorporation into tumor-infiltrating MDSCs. Hence, pharmacological inhibition of this pathway suppresses the activity of MDSCs, delays tumor growth and enhances the antitumor effect of therapies such as adoptive-T-cell-transfer therapy or low-dose chemotherapy [50] (Figure 4).

### 4.3. Modulating T Cells

Blockade of PD-L1 in gastric adenocarcinomas has been observed to influence lipid metabolism by increasing FAPB4/5 expression in CD8+ tissue-resident memory T cells, thereby improving both lipid uptake and their survival in vitro and in vivo. Another study linked PD-L1 to metabolic reprogramming favoring FAO [121]. In addition, it appears that obesity can promote tumor growth by remodeling CD8^+^ lipid metabolism through a leptin–STAT3–FAO pathway [122]. 

Recent studies have shown that metabolic reprogramming can occur in tumor cells and immune cells. However, the effects on T cells of traditional lipid metabolism drugs, such as statins, remains controversial. On the other hand, statins inhibit checkpoint expression on T lymphocytes [123]. The metabolic pathway of mevalonate kinase (MVK) is involved in the synthesis of cholesterol. MVK is also crucial for the activation of T cells in an AKT/mTOR signaling-dependent manner [124]; the latter represents one of the main pathways of cell metabolism and proliferation-regulation, making it broadly targeted in oncology or in autoimmune diseases [125]. In a microenvironment with hypoglycemia and hypoxia, most T cells are inactivated through overexpression of checkpoint (such as PD-1 and LAG-3), and the free FAs around them are dramatically increased. Fenofibrate, used in the treatment of patients with cholesterol excess, can increase FAO of T cells by activating PPARα, thus reversing the inhibitory effect of the tumor microenvironment [126]. Moreover, FAO inhibition can also impair differentiation and activity of Treg cells, angiogenesis, and cancer-cell metabolism [32]. Interestingly, by dispensing with the use of drugs it would be possible to modulate the differentiation of CD4 T cells. Indeed, it has been shown in mice that a diet rich in DHA, a polyunsaturated fatty acid, makes it possible to limit Th17 cell differentiation through PPARγ, a nuclear receptor that regulates lipid absorption and intracellular metabolism. This diet slowed tumor growth in mice [127] (Figure 4).

## 5. Conclusions

Lipid metabolism plays an important role in tumor growth and survival. Thereby, the implication of this metabolic switch in the failure of anticancer therapies has emerged in the last decade. Indeed, numerous studies have highlighted the changes in differentiation, proliferation and pro- or antitumor activity of immune cells (innate or adaptive) in response to lipid changes. Many questions are unanswered, and the impact of lipid metabolism on immune cells remains an emerging area of study. Modulation of the antitumor activity of cancer stem cells (CSCs) could be one of these targets. Indeed, a recent article highlights that the knockdown of Arf-1-mediated metabolism pathway mediates CSC death and induces a tumor-specific immune response [128]. Modulation of ferroptosis, a novel regulated-cell-death (RCD), could also be another target. RCD, mediated by iron accumulation and unrestricted lipid peroxidation, induces the regulation of cellular immune response via the release of damage-associated molecular patterns (DAMPs) and lipid metabolites. During this new form of immunogenic cell death (ICD), FAs are catalyzed into various PUFAs, such as AA, which are then esterified into phosphatidylethanolamines or transformed into lipid metabolites [129]. A new ferroptosis therapy has been proposed with the tumor-targeted delivery of doxorubicin (DOX)-Fe^2+^ [130]. Finally, modulation of the metabolism of immune cells other than those previously described could also a target for therapy. Thus, invariant natural killer T cells (iNKT), known for their important role in antitumor immune responses, are very sensitive to impaired metabolism. It has been shown that combined treatment with a drug used to treat type 2 diabetes (pioglitazone) and apha-galactosylceramide induces lipid synthesis via PPARγ activation and significantly enhances iNKT cell-mediated antitumor immune response [131]. The discovery of all these mechanisms should provide new potential drug targets to improve cancer treatment. The challenge is to identify specific lipid species or receptors that are involved in tumors but do not play an essential role in lipid metabolism in normal tissues. 

## Figures and Tables

**Figure 1 cancers-14-01850-f001:**
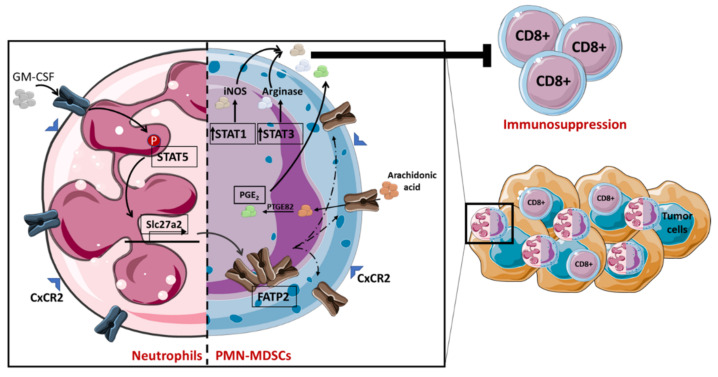
**Neutrophils reprogramming to PMN-MDSCs.** GM-CSF attaches to the neutrophil receptor, which undergoes gene rearrangement via STAT5 phosphorylation, leading to *scl27a2* gene transcription and FATP2 expression. This mechanism leads to the conversion of neutrophils to PMN-MDSCs. Once the reprogramming is complete and FATP2 is attached to the membrane, the PMN-MDSCs will absorb the arachidonic acid, which will be responsible for the synthesis and the secretion of PGE2. The latter, as well as iNOS (produced in response to STAT1 signaling pathway) and arginase (produced in response to STAT3 signaling pathway) will induce the suppression of CD8 T lymphocytes and therefore lead to the proliferation of cancer cells.

**Figure 2 cancers-14-01850-f002:**
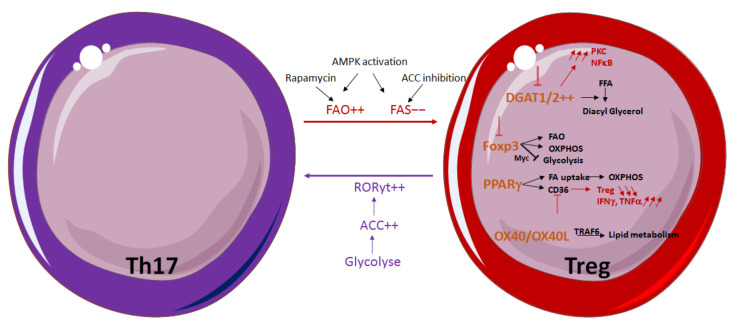
**Fatty acid metabolism is involved in Th17/Treg balance.** Th17/Treg metabolism is regulated by glycolysis and FAO. In Treg, Foxp3 favors FAS and OXPHOS but decreases glycolysis. DGAT1/2, PPARγ and OX40/OX40L are involved in FA metabolism in Tregs.

**Figure 3 cancers-14-01850-f003:**
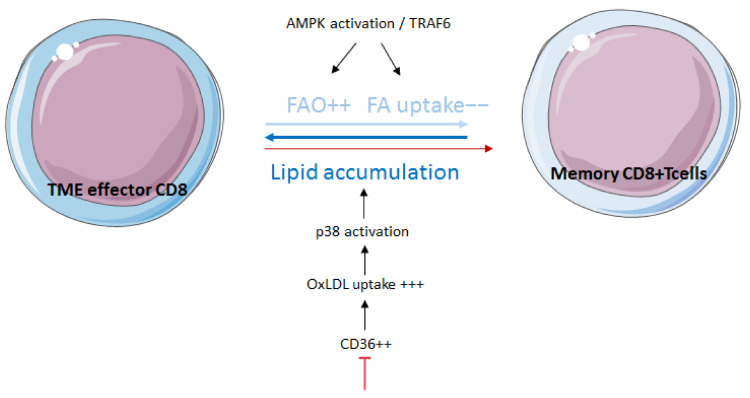
**Fatty acid metabolism modulates effector and memory CD8 T cells.** FAO is essential for memory CD8+ cell generation through the AMPK/TRAF6 pathway. FAO in memory CD8+ cells is mainly fueled through FA uptake by CD36.

**Figure 4 cancers-14-01850-f004:**
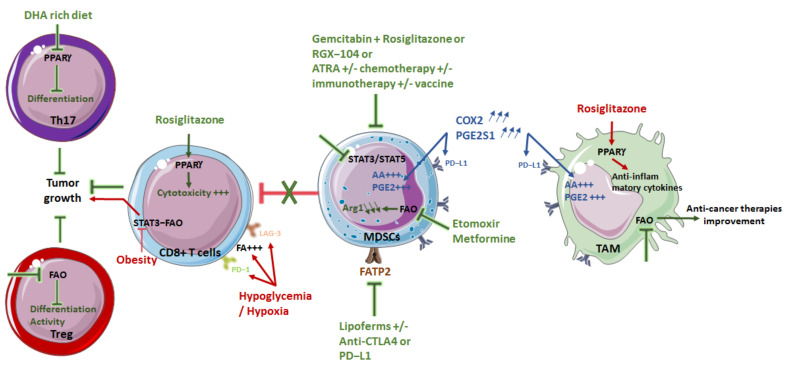
**Targeting immune cell lipid metabolism as a therapeutic strategy.** PPAR agonist (rosiglitazone) with or without chemotherapy induces an increase of anti-inflammatory cytokines in M2 macrophages, an increase in CD8+ T-cell cytotoxicity as well as inhibition of MDSC immunosuppressive function. In contrast, inhibition of PPAR, following a DHA-rich diet, inhibits Th17 differentiation. All of these changes lead to a decrease of tumor progression. Overexpression of COX2 and PGE2S1 are also capable of targeting tumor growth by increasing arachidonic acid (AA), PGE2 production and PD-L1 expression both in MDSCs and in TAMS. FAO inhibitors also induce an alteration of MDSCs, TAM and Treg differentiation and function, leading to improvement of anti-cancer therapies. Lipoferms, an inhibitor of FATP2, with or without immunotherapies targeting CTLA4 or PD-L1, as well as inhibitors of STAT3/STAT5 induce a strong inhibition of MDSC function and decreased tumor growth. Conversely, inhibition of the leptin–STAT3–FAO pathway on CD8+ T cells, which occurrs in obese individuals, promotes tumor growth. Similarly, hypoglycemia and hypoxia lead to CD8+ T-cell inhibition through overexpression of PD-1 and LAG-3, which increases free FA around them.

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
