# Peer review of "Impact of Lipid Metabolism on Antitumor Immune Response"

_cancers, 2022, doi:10.3390/cancers14071850_

Round 1
Reviewer 1 Report
The review „Impact of lipid metabolism on anti-tumor immune response“ by Mabrouk et al. sums up the current data on the influence of metabolic, overall of lipid and fatty acid metabolism, on the anti-tumor immune response. Actually, this is a current topic of interest to a broad readership, since it bridges to major topics in oncology, the metabolic and immune regulation. The review is very comprehensive and well writen. Some minor spelling errors should be corrected.
In order to improve the clarity of the main issues, I would ask for an overview (e.g., a new table or figure) in which the author’s summarize clinical and or pre-clinical data with references. This may be ordered e.g. by specific drugs (such as antidiabetic drugs, immunosuppresives, chemotherapies or their interactions) or molecular targets (such as PPAR, IL, PGE, etc.). Doing so, the review may further lead to a mechanistic understanding, going beyond cell type specific regulations.
Reviewer 2 Report
Journal Cancers (I S S N 2072-6694)
Manuscript ID cancers-1660484
Type Review
Title Impact of lipid metabolism on anti-tumor immune response
Authors Nesrine Mabrouk , Baptiste Lecoeur , Ali Bettaieb , Catherine Paul , Frederique VEGRAN *
Section Tumor Microenvironment
Collection Deciphering the Crosstalk between Tumor Cells and Their Microenvironment: From Molecular Aspects to Therapeutic Implications
Date: 24 MARCH
PEER REVIEW R E P O R T:
In this manuscript, Mabrouk et. al have explored the role of lipid metabolism in anti-tumor immunity. Although, the central idea of the manuscript is interesting, it needs to be improved. Below are my comments:
- Authors can improve the introduction as it seems abrupt. For example something like this can be added from literature, 'Immunosurveillance is a term used to describe the processes by which cells of the immune system look for and recognise foreign pathogens, such as bacteria and viruses, or pre-cancerous and cancerous cells in the body. https://www.nature.com/subjects/immunosurveillance.'
- Before shifting to metabolic reprogramming, authors can build an introduction to the general concepts of metabolic plasticity. Refer second paragraph 'Metabolic reprogramming of immune cells is being defined as the new hallmark of 41 cancer, which changes the functionality of immune cells by controlling transcriptional and 42 posttranscriptional events that are essential for the activation of immune cells[2]. Specific 43 metabolic pathway alterations affect immune cell functions'
- Authors can explore the following reference 'Fendt, Sarah-Maria, Christian Frezza, and Ayelet Erez. "Targeting metabolic plasticity and flexibility dynamics for cancer therapy." Cancer discovery 10, no. 12 (2020): 1797-1807.'
- Similarly, authors need to expand the relevance of 'innate and adaptive immune cells in TME' and add some references. Refer the paragraph 'Innate and adaptive immune cells are involved in the TME and affect tumor growth 49 by either triggering an inflammatory response to participate in the elimination of cancer 50 cells or by displaying a pro tumor immunosuppressive activity.'
- Authors can discuss the distinction/variation of macrophage polarization in mouse compared to humans. Macrophage polarization in humans although visible is not as clear as mouse but is rather the continuum. Refer to manuscript 'The M1 macrophages are classically 66 activated. They have proinflammatory features, and occur in response to bacterial infec-67 tion products (e.g. lipopolysaccharide (LPS) and interferon gamma (IFNg)). This subtype 68 is also defined for its anti-tumor properties. The M2 macrophages, alternatively activated 69 in response to products associated with parasitic infection (e.g. Schistosoma egg antigen, 70 interleukins (IL)-4 and -13), have anti-parasitic activities, tissue repair and remodeling 71 features [9,10]. '
- For the above section, authors can take help of following references: Vijayan, Vijith, Pooja Pradhan, Laura Braud, Heiko R. Fuchs, Faikah Gueler, Roberto Motterlini, Roberta Foresti, and Stephan Immenschuh. "Human and murine macrophages exhibit differential metabolic responses to lipopolysaccharide-A divergent role for glycolysis." Redox biology 22 (2019): 101147.
- Murray, P.J. and Wynn, T.A., 2011. Obstacles and opportunities for understanding macrophage polarization. Journal of leukocyte biology, 89(4), pp.557-563.
- Murray, P.J., Allen, J.E., Biswas, S.K., Fisher, E.A., Gilroy, D.W., Goerdt, S., Gordon, S., Hamilton, J.A., Ivashkiv, L.B., Lawrence, T. and Locati, M., 2014. Macrophage activation and polarization: nomenclature and experimental guidelines. Immunity, 41(1), pp.14-20.
- Authors should rephrase this sentence 'The modification of the lipid metabolism is one of the characteristics thanks to which 535 the tumor can grow and survive. '
- In a recent study, 'Cancer stem cells (CSCs) may be responsible for treatment resistance, tumor metastasis, and disease recurrence. Here we demonstrate that the Arf1-mediated lipid metabolism sustains cells enriched with CSCs and its ablation induces anti-tumor immune responses in mice. Notably, Arf1 ablation in cancer cells induces mitochondrial defects, endoplasmic-reticulum stress, and the release of damage-associated molecular patterns (DAMPs), which recruit and activate dendritic cells (DCs) at tumor site' Refer 'Wang, Guohao, Junji Xu, Jiangsha Zhao, Weiqin Yin, Dayong Liu, WanJun Chen, and Steven X. Hou. "Arf1-mediated lipid metabolism sustains cancer cells and its ablation induces anti-tumor immune responses in mice." Nature communications 11, no. 1 (2020): 1-16.
- Also, ferroptosis has emerged as an important player. 'Ferroptosis is a newly described RCD, which is driven by iron accumulation and unrestricted lipid peroxidation. 'Shi, Lei, Yingqi Liu, Menghuan Li, and Zhong Luo. "Emerging roles of ferroptosis in the tumor immune landscape: from danger signals to anti‐tumor immunity." The FEBS Journal (2021).
- Authors can also discuss the importance of iNKT cells 'Fu, Sicheng, Kaixin He, Chenxi Tian, Hua Sun, Chenwen Zhu, Shiyu Bai, Jiwei Liu et al. "Impaired lipid biosynthesis hinders anti-tumor efficacy of intratumoral iNKT cells." Nature communications 11, no. 1 (2020): 1-15.'
Round 2
Reviewer 2 Report
The authors have done a tremendous job by compiling this relevant and timely review. The revised version has added new segments to the introduction, immune part, and conclusion section. It will make this manuscript informative for researchers from diverse perspectives.